# The Importance of Thermal Treatment on Wet-Kneaded Silica–Magnesia Catalyst and Lebedev Ethanol-to-Butadiene Process

**DOI:** 10.3390/nano11030579

**Published:** 2021-02-26

**Authors:** Sang-Ho Chung, Adrian Ramirez, Tuiana Shoinkhorova, Ildar Mukhambetov, Edy Abou-Hamad, Selevedin Telalovic, Jorge Gascon, Javier Ruiz-Martínez

**Affiliations:** 1KAUST Catalysis Center, King Abdullah University of Science and Technology, Catalysis, Nanomaterials, and Spectroscopy (CNS), Thuwal 23955, Saudi Arabia; ildar.mukhambetov@kaust.edu.sa; 2KAUST Catalysis Center, King Abdullah University of Science and Technology, Advanced Catalytic Materials (ACM), Thuwal 23955, Saudi Arabia; adrian.galilea@kaust.edu.sa (A.R.); tuiana.shoinkhorova@kaust.edu.sa (T.S.); selvedin.telalovic@kaust.edu.sa (S.T.); jorge.gascon@kaust.edu.sa (J.G.); 3KAUST Core Labs, King Abdullah University of Science and Technology, Thuwal 23955, Saudi Arabia; edy.abouhamad@kaust.edu.sa

**Keywords:** ethanol, butadiene, Lebedev process, wet-kneading, silica–magnesia, magnesium silicate

## Abstract

The Lebedev process, in which ethanol is catalytically converted into 1,3-butadiene, is an alternative process for the production of this commodity chemical. Silica–magnesia (SiO_2_–MgO) is a benchmark catalyst for the Lebedev process. Among the different preparation methods, the SiO_2_–MgO catalysts prepared by wet-kneading typically perform best owing to the surface magnesium silicates formed during wet-kneading. Although the thermal treatment is of pivotal importance as a last step in the catalyst preparation, the effect of the calcination temperature of the wet-kneaded SiO_2_–MgO on the Lebedev process has not been clarified yet. Here, we prepared and characterized in detail a series of wet-kneaded SiO_2_–MgO catalysts using varying calcination temperatures. We find that the thermal treatment largely influences the type of magnesium silicates, which have different catalytic properties. Our results suggest that the structurally ill-defined amorphous magnesium silicates and lizardite are responsible for the production of ethylene. Further, we argue that forsterite, which has been conventionally considered detrimental for the formation of ethylene, favors the formation of butadiene, especially when combined with stevensite.

## 1. Introduction

1,3-Butadiene (butadiene) is an essential C4 monomer in the polymer industry for styrene–butadiene rubber (SBR), acrylonitrile–butadiene–styrene rubber (ABS), and poly-butadiene–styrene (PBS). Butadiene is currently produced as a byproduct in naphtha steam crackers [1]. Thus, there is a strong dependence on the price of ethylene (the main product of naphtha steam cracking) [2,3,4,5], with the recent increase in shale gas usage potentially leading to butadiene shortage in the global chemical market [1,6]. There has been a growing need for on-purpose, sustainable processes for butadiene. The Lebedev process is one such promising candidates owing to its innate “target-specific” for butadiene from ethanol. In addition, given the recent increased availability of bio-ethanol [7], a sustainable butadiene process is particularly attractive. 

Although the actual reaction mechanism of the Lebedev process is still under discussion, the aldol condensation mechanism proposed by Toussaint et al. is one of the most plausible ones [2,8]: ethanol is first converted into acetaldehyde by dehydrogenation and aldol condensation between two acetaldehyde forms of 3-hydroxybutanal; subsequent dehydration and hydrogenation yields crotonaldehyde and crotonyl alcohol, respectively; and finally, butadiene is obtained by dehydration (Scheme 1). These multistep conversions require a multifunctional catalyst, and the interplay of acid/base and redox properties of the catalyst plays a crucial role in achieving a high butadiene yield. 

Various metal-promoted catalysts were recently studied in the Lebedev process such as M–Ta/BEA (M = Ag, Cu, and Zn) [9], Zn–Y/BEA (5% Zn and 5% Y) [10], Cu or Zn/MgO–SiO_2_ [11], Hf–Zn/SiO_2_ [12], and Ga/Mg–SiO_2_ (5% Ga and Si/Mg = 1) [13]. Ta, Y, and Mg can alter the acid/base properties, while the second metals such as Ag, Cu, Zn, Hf, and Ga improve the redox properties of the catalysts. Although promotion of the redox function by a second metal can enhance the dehydrogenation activity and butadiene selectivity, metal-promoted catalysts often show fast deactivation within 24 h time-on-stream due to the considerable amount of carbon deposition and/or agglomeration of the metal species [14]. 

SiO_2_–MgO catalysts are benchmark catalysts for the Lebedev process [15,16,17]. The preparation method of the SiO_2_–MgO catalyst is crucial for its catalytic performance. For example, the physical mixture of SiO_2_ and MgO shows much worse catalytic performance when compared to the coprecipitated SiO_2_–MgO [15]. Impregnation of magnesium species on mesoporous silica (MgO promoted SBA-15) efficiently converts ethanol into acetaldehyde due to the formation of moderate-strength Mg^2+^–O^2−^ Lewis acid–base site pairs at the expense of strongly basic MgO sites, stabilizing the transition state of adsorbed ethanol [17]. However, these MgO-promoted SBA-15 catalysts could not yield butadiene.

SiO_2_–MgO catalysts prepared by wet-kneading showed better catalytic performance, not only in terms of ethanol conversion but also in butadiene selectivity [15,18]. During wet-kneading, various types of magnesium silicates are formed by partial dissolution of individual SiO_2_ and Mg(OH)_2_ components (i.e., [SiO_2_(OH)_2_]^2−^ and [Mg(H_2_O)_6_]^2+^) in a basic wet-kneading solution (pH > 9). The opposite surface charge of wet-kneading precursors have the dissolved ions cross-deposited on the surfaces [19]. As a result, a minimum of five different magnesium silicate polymorphs are formed, such as talc (Mg_3_Si_4_O_10_(OH)_2_), lizardite (Mg_3_Si_2_O_5_(OH)_4_), and stevensite (Mg_x_(Mg_3−x_◻_x_)Si_4_O_10_(OH)_2_, ◻ = defect site) as hydrous magnesium silicates, and forsterite (Mg_2_SiO_4_) and an intermediate species between forsterite and enstatite (MgSiO_3_) as anhydrous ones [19]. The surface magnesium silicates of wet-kneaded SiO_2_–MgO attain the balanced acid–base and redox properties with altered Si^4+^–O^2−^ and Mg^2+^–O^2−^ sites and, correspondingly, show its greater activity in the Lebedev process. 

Due to the structural complexity of wet-kneaded SiO_2_–MgO, the role of magnesium silicate on the Lebedev process has not been clarified yet [6]. Moreover, the characterization of wet-kneaded SiO_2_–MgO is challenging because cross-deposition only occurs on precursor surfaces, forming only a few layers of mixed-phase magnesium silicates. For instance, Janssens et al. reported that surface silanol groups in SiO_2_ are responsible for the acidic nature and dehydration activity [20]. Chung et al. reported that amorphous magnesium silicates are responsible for promoting ethanol dehydration to ethylene and layered Mg–Si promotes butadiene formation [19]. In addition, Ochoa et al. observed that the formation of forsterite (Mg_2_SiO_4_) by the sol-gel method had the strongest acidity among the series of sol-gel SiO_2_–MgO catalysts, which is detrimental to butadiene selectivity, while favors ethanol dehydration to ethylene [21]. 

Here, we aim to understand the individual role of magnesium silicate in the wet-kneaded SiO_2_–MgO catalyst in the Lebedev process. Varying the calcination temperature, a series of catalysts was prepared, and various characterization were used to study the nature and amount of magnesium silicates, which are largely influenced by thermal treatment. The catalytic performance of the prepared catalysts was evaluated at different ethanol feed rates for proper comparison. We expect that our results may enable further optimization of the wet-kneaded SiO_2_–MgO catalyst and the ethanol-to-butadiene process.

## 2. Materials and Methods 

### 2.1. Materials

Tetraethyl orthosilicate (≥99.0%, Sigma-Aldrich, Steinheim, Germany), ammonium hydroxide (99.99%, 25% NH_3_, Alfa Aesar, Lancashire, UK), ethanol absolute (≥99.8%, Fisher Scientific, Fair Lawn, NJ, USA), magnesium nitrate hexahydrate (≥99.0%, Sigma-Aldrich, Steinheim, Germany), and sodium hydroxide (≥97.0%, Sigma-Aldrich, Steinheim, Germany) were used as received without further purification. 

### 2.2. Catalyst Preparation

The wet-kneaded SiO_2_–MgO catalyst was prepared from SiO_2_ and Mg(OH)_2_ precursors according to a published procedure [19]. The Stöber SiO_2_ was prepared by hydrolysis of tetraethyl orthosilicate in ethanol/ammonium hydroxide solution. After 15 h of aging, the solid SiO_2_ nanoparticles were obtained using a rotary evaporator at reduced pressure. The Mg(OH)_2_ was synthesized adding 0.4 M NaOH aqueous solution dropwise to 0.2 M Mg(NO_3_)_2_ aqueous solution until the pH reached 12. The precipitated Mg(OH)_2_ particles were separated by centrifugation and washed multiple times with deionized (DI) water. The SiO_2_ and Mg(OH)_2_ precursors were dried at 120 °C overnight. The precursors were wet-kneaded in DI water at room temperature for 4 h. The resultant solid was recovered by centrifugation and dried at 120 °C overnight. The sample is named WK-dried. The WK-dried sample was then finally thermally treated at different temperatures from 500 to 900 °C for 5 h and denoted with the calcination temperature. 

### 2.3. Characterization

Scanning electron microscopy (SEM) images of wet-kneading precursors were taken at FEI Teneo VS microscope (FEI Company, Hillsboro, OR, USA). The electron was accelerated at 5 kV, and the images were acquired at around 2 mm working distance. 

Powder X-ray diffraction (PXRD) patterns of the materials were recorded on a Bruker D8 Advance X-ray diffractometer (Bruker, Billerica, MA, USA) equipped with a LynxEye position-sensitive detector in 1D mode. The measurements were acquired by using monochromatic Cu–Kα (λ = 1.5418 Å) radiation operated at 40.0 kV. The scan ranged from 2θ = 5 to 80° with a step size of 0.021° and a scan speed of 1.0 s per step. The crystalline phases were identified with the help of the PDF-4+ (2019) crystal database. The crystallite size of the catalysts were calculated using the Scherrer equation with a shape factor of 0.89 and an instrument width of 0.05. 

The analysis of textural properties was achieved by nitrogen adsorption at −196 °C using Micromeritics ASAP 2420 high-throughput analysis system. Prior to the measurement, the samples were outgassed at 200 °C under vacuum for 8 h. Specific surface areas were estimated according to the Brunauer–Emmett–Teller (BET) method in the relative pressure range (p/p_0_) of 0.05–0.95. The pore size distribution was analyzed by using the Barrett–Joyner–Halenda (BJH) method applied to the desorption branch. 

One-dimensional (1D) ^1^H spin-echo (SE), ^29^Si direct excitation (DE), ^1^H–^29^Si cross-polarization (CP), and two-dimensional (2D) ^1^H–^29^Si heteronuclear chemical shift correlation (HETCOR) magic angle spinning (MAS) solid-state nuclear magnetic resonance (NMR) experiments were recorded on a Bruker AVANCE III spectrometer (Bruker BioSpin, Rheinstetten, Germany) operating at a ^1^H Larmor frequency of 400 MHz. One-dimensional and two-dimensional NMR spectra (manufacturer, city, (state or province), country) were recorded using a 4 mm Bruker triple-resonance and a double-resonance MAS probe, respectively. Dry nitrogen gas was utilized for sample spinning at 12 kHz. For ^1^H SE MAS NMR, a 50 kHz π/2 radio frequency (RF) pulse was used, followed by π pulse for refocusing, and accumulated for 64 scans. For 1D ^29^Si DE, the spectra were recorded using 125 kHz π/2 pulse with a long recycle delay of 20 s for complete relaxation of ^29^Si nuclei. For ^1^H–^29^Si CP, the spectra were recorded using a 4 s recycle delay, 20 ms acquisition time, and accumulation of 22,528 scans. Cross-polarization was achieved using a 85 kHz ^29^Si field and 38 kHz 70–100% ramped ^1^H field for 3 ms. For 1D NMR, the ^1^H and ^29^Si spectra were processed using 40 and 50 Hz line-broadening, respectively. The 2D frequency-switched Lee–Goldberg ^1^H–^29^Si HETCOR spectra were acquired with a 12 kHz MAS frequency [22]. The pulse sequence started with a proton pulse, followed by a train of frequency and phase-switched LG pulses (t1 evolution period), and the ramped amplitude CP to ^29^Si. During the CP step, ^1^H CP spin lock pulses centered at 38 kHz were linearly ramped from 75 to 100% and the ^29^Si RF field was matched to obtain an optimal signal. The CP contact time was 3 ms. During the acquisition (t2), the protons were decoupled using a SPINAL-64 sequence. In 2D, linear prediction was used to double the points in the indirect dimension and the exponential multiply window function was applied to both time dimensions. All spectra were referenced to hexamethyl cyclosilane. 

Fourier-transform infrared (FTIR) spectra (Thermo Fisher Scientific, Waltham, MA, USA) for pyridine-adsorbed catalysts were collected using a Nicolet 6700 instrument using deuterated triglycine sulfate as the detector with a resolution of 4 cm^−1^ and 64 scans. The self-supported catalyst pellets (ca. 50 mg of catalyst) were prepared by using a pelletizer with 5 ton pressure. The degassing and IR measurements of the pellet was performed under vacuum using a customized glass setup connected to a Pfeiffer HiCUBE High Vacuum system. The pellet was heated to 90 °C for 1 h and to 400 °C for 3 h, with a heating ramp of 10 °C/min. After cooling down to room temperature, pyridine vapor was introduced to the pellet for 20 min and the glass chamber was evacuated for 30 min to remove physisorbed pyridine. The pellets were heated to 150 °C for 17 min under vacuum and then IR spectra were collected. 

Temperature-programed desorption of ammonia (NH_3_–TPD) and carbon dioxide (CO_2_–TPD) were performed on a Micrometrics ASAP 2920 unit (Micromeritics, Norcross, GA, USA). First, the sample (ca. 100 mg) in a quartz reactor was heated (10 °C/min) in He flow for drying at 350 °C for 30 min. Subsequently, for NH_3_–TPD, the sample was cooled down to 40 °C and the flow was switched to NH_3_ (10 vol.% in He) for 15 min. For CO_2_–TPD, the sample was cooled down to 50 °C, and pure CO_2_, which was previously passed through moisture filter, was flown for 15 min. Then, the flow was switch to He and maintained at 40 °C for NH_3_ (50 °C for CO_2_) for 15 min to remove physisorbed species on the catalyst surface. Finally, the sample was heated to 500 °C with a ramping rate of 5 °C/min (to 700 °C with the rate of 10 °C/min for CO_2_) under He, and the desorption of NH_3_ and CO_2_ was detected by using a thermal conductivity detector (TCD) and a mass spectrometry detector. The total amount of the adsorbed species was quantified from the preliminary calibrated Cirrus 2 mass spectrometer (MKS Spectra Product, Andover, MA, USA) using m/z = 16 and 28 signals for NH_3_ and CO_2_, respectively. 

### 2.4. Activity Test

Catalytic experiments were carried out in a 4-channel Flowrence XD high-throughput reactor system (Avantium, Amsterdam, The Netherlands). The reaction tests were carried out at 425 °C and ambient pressure. The catalyst bed was diluted with silicon carbide (SiC) to decrease the effect of axial dispersion and to improve heat conduction in the bed. Typically, 50 µL of catalyst was mixed with 200 µL of SiC (grit 46) and placed in a quartz tube with an internal diameter of 2.3 mm. A high-performance liquid chromatography pump was used to feed liquid ethanol to the reactor system, and the liquid-hourly space velocity (LHSV, liquid flow volume per hour and per catalyst volume) varied from 0.5 to 1.5 h^−1^ in nitrogen as a carrier gas. The reaction products were analyzed by gas chromatography (GC) in an Agilent 7890B with three detectors: two flame ionization detectors (FIDs) and one TCD. The TCD channel had a PoraPLOT Q GC column as a backflush column; a Hayesep Q column for separation of CO_2_; and a Molsieve as an analytical column for the separation of He, H_2_, N_2_, CH_4_, and CO. All other compounds (water, hydrocarbons, and oxygenates) were backflushed. The FID was equipped with a 10 m precolumn with a wax stationary phase. The separation of C_1_–C_5_ hydrocarbons was carried out on a 30 m Gaspro stationary phase. The separation of ethanol, acetaldehyde, and the rest of the oxygenates was carried out on a 30 m wax stationary phase. 

The conversion of ethanol (X) and product selectivities (S_i_) were calculated based on the following formulas:(1)$$X\;=\;\frac{C_{{\mathrm{EtOH}}_{\mathrm{in}}}{\;-\;C}_{{\mathrm{EtOH}}_{\mathrm{out}\;}}}{C_{{\mathrm{EtOH}}_{\mathrm{in}}}}\;\times \;100$$
(2)$$S_{i}\;=\;\frac{{i\;\times \;C}_{i}}{C_{{\mathrm{EtOH}}_{\mathrm{in}}}{\;-\;C}_{{\mathrm{EtOH}}_{\mathrm{out}}}}\;\times \;100$$
where $C_{{\mathrm{EtOH}}_{\mathrm{in}}}$ and $C_{{\mathrm{EtOH}}_{\mathrm{out}}}$ are the concentrations determined by GC analysis of ethanol in the blank and in the reactor.

## 3. Results and Discussion

### 3.1. Effect of Thermal Treatment on Wet-Kneaded SiO_2_–MgO Catalyst 

The size and morphology of the precursors (SiO_2_ and Mg(OH)_2_) heavily influence the physicochemical properties of wet-kneaded SiO_2_–MgO catalysts and, correspondingly, the catalytic activity of the Lebedev process [19,23]. In this study, we utilized the best precursors based on our previous result [19] to induce an optimal intimate contact, which are spherical SiO_2_ particles of 45 nm in diameter and platelet-like hexagonal Mg(OH)_2_ prisms of around 135 and 25 nm in length and height, respectively. The SEM images of the catalyst precursors can be found in Appendix A.

Figure 1a shows the X-ray diffraction patterns of the prepared wet-kneaded SiO_2_–MgO catalysts calcined at different temperatures. A typical hexagonal Mg(OH)_2_ brucite phase with P–3m1 space group symmetry (PDF# 00–044–1482) was observed in the WK-dried sample (Figure 1a). After calcination at 500 °C, the brucite (001) phase dehydroxylated to form a MgO cubic periclase (111) phase (Fm–3m, PDF# 04–010–4039) [24], either by the migration of Mg^2+^ [25] or by the oxygen layers slip [26]. Crystal-size analysis using the Scherrer equation shows 21 nm size brucite crystals disappearing and forming new and smaller (8.5 nm) periclase crystals (Figure 1b and Table 1). Once the periclase phase formed, these crystallites gradually agglomerated with increasing calcination temperature. 

Notably, for the calcined catalysts at higher temperatures (WK-800 and WK-900), the development of crystalline magnesium silicates, more specifically, orthorhombic Mg_2_SiO_4_ forsterite (Pbnm, PDF# 01–175–6661) was observed. The characteristic diffraction pattern of forsterite became more intense for the WK-900 catalyst. It is worth noting that other anhydrous magnesium silicates such as enstatite (MgSiO_3_) and its polymorphs were not observed. In addition, all samples showed a broad contribution in the region of 20–30° 2θ. This strongly suggests that the wet-kneaded SiO_2_–MgO catalysts still preserve an amorphous SiO_2_ phase in the inner core of spherical SiO_2_ particles, i.e., wet-kneading modifies only the precursor surfaces. 

The textural properties of the catalysts were investigated by N_2_ adsorption at −196 °C. The wet-kneaded SiO_2_–MgO catalysts displayed type IV adsorption isotherms, with an H3 loop characteristic of layered materials containing wide mesopores (Figure 2). Notably, variation in the calcination temperature shows a volcano-shaped trend of the BET surface area of the samples, showing a maximum with the WK-500 catalyst. Compared to WK-dried, the surface area of WK-500 increased by about 33%. The dehydration of brucite crystals occurred while the water molecules escaped from the crystalline lattices. Owing to the inhomogeneous dehydration of brucite [25,26], pores of sizes around 3.5 nm were generated after calcination at 500 °C (Appendix A). The surface area and pore volume of the catalysts decreased significantly at higher calcination temperatures (Table 1). WK-800 and -900 exhibited type IVb isotherms [27], suggesting not only small pores, which were formed by brucite dehydration, but also that the one end of the interparticle pores (ca. 10 nm) are partially blocked by calcination at 800–900 °C, forming forsterite on the catalyst surface (Appendix A). 

Further insights into the structure of the catalysts were investigated in detail by solid-state NMR spectroscopy. This tool was proven to play a pivotal role in the understanding of magnesium silicates formed by wet-kneading [19,23]. Although ^1^H NMR could provide structural insights of the hydroxyl groups on SiO_2_ [28], MgO [29], and pure magnesium silicates [30,31,32], ^1^H spectra of wet-kneaded SiO_2_–MgO catalysts has not been reported so far, probably due to the massive signals from adsorbed water and the complexity of the various proton signals. Here, we provide the ^1^H MAS NMR spectra of the wet-kneaded SiO_2_–MgO catalysts calcined at different temperatures (Figure 3). For a better understanding, the spectra of the individual SiO_2_ and MgO are also summarized in Appendix A. Although we could not completely average the strong dipolar coupling of water molecules, (at least) two broad resonances at around 4 and 5 ppm were observed, showing the heterogeneity of adsorption sites for water on the catalysts. It is known that the chemical shift in adsorbed water on the catalyst surface is influenced by the hydrogen bond strength between water and the surface hydroxyl groups [28,29]. The chemical shift and intensity of the resonance of physisorbed water can also be affected by the local electric field from surface polarity [30,33]. Notably, WK-800 still has a considerable amount of water on the surface, suggesting its hydrous features. On the contrary, WK-900 showed its anhydrous features: the resonance at around 5 ppm disappeared, and only a small amount of surface water was observed at 3.7 ppm due to the formation of nonpolar and rigid O–Mg–O layers of forsterite [34,35], with a small contribution of Q4 siloxane groups on the SiO_2_ [28]. 

This leaves three signals at around 2.0, 1.1, and 0.6 ppm to be assigned. The regions from 0 to 2 ppm can provide insights into the actual structural hydroxyl groups but it is not straightforward due to extensive overlaps. For example, all pure precursors exhibit features at around 2 ppm (Appendix A). Dumas et al. indicated that synthetic talc displays a ^1^H resonance of silanol on lateral sheet edges at 1.8 ppm [30], while Poirer et al. assigned this peak to residual sodium acetate during the synthesis [36]. For WK-dried and WK-500, the ^1^H resonance at around 0.6 ppm is one of the predominant resonances with small features of 1.1 ppm. From its asymmetric features, it is expected to be overlapped with several features of hydrous magnesium silicates [19], as well as MgO (Appendix A). Interestingly, upon calcination at higher temperature, the signal at 0.6 ppm shifted downfield by 0.3 ppm and became more resolved, suggesting that isolated hydroxyl groups become more structurally ordered. Because of the complexity of these spectra, assignment of the remaining contributions will be discussed with the 2D ^1^H–^29^Si HETCOR results below. 

The ^29^Si and ^1^H–^29^Si CP MAS NMR spectra provide additional information on the nature of magnesium silicates on wet-kneaded SiO_2_–MgO catalysts (Figure 4). In the ^29^Si direct excitation (DE) spectra (Figure 4a), the resonance at −109.8 ppm, ascribed to Q4 siloxane groups, is the most intense for all catalysts, suggesting that the inner core of spherical SiO_2_ is maintained during wet-kneading, in agreement with our XRD results. Compared to pure SiO_2_ nanoparticles, the ^29^Si resonances of the catalysts are asymmetrically broadened, especially downfield, showing an alteration of the chemical environment of silicon species by wet-kneading (Table 2). After calcination, a large portion of silicon species at around −100.0 ppm (e.g., the so-called simple silanol groups, (SiO)_3_–Si–OH)) were transformed to magnesium silicates (WK-dried and WK-500 in Figure 4a). A broad signal was observed at −79.0 ppm, attributed to a combinations of a structurally ill-defined (amorphous) hydrous magnesium silicate (in the region of −85 to −92 ppm) [37,38] and intermediate species between forsterite and enstatite (−77.6 ppm) [39]. In addition, a resonance of poorly crystalline lizardite (−93.8 ppm) [19,40] was clearly observed as the calcination temperature increased. The intensities of ^29^Si resonances in the region −70 to −95 ppm increase with calcination temperatures up to 800 °C. For WK-900, the formation of forsterite (−61.3 ppm) was clearly observed [41], at the expense of the signals of magnesium silicates in the region of −70 to −100 ppm. WK-900 displayed a broaden Q4 resonance despite the highest temperature calcination, probably due to the formation of additional SiO_2_ phases by the dehydroxylation of the magnesium silicates.

With ^1^H–^29^Si CP MAS experiments, we could obtain more resolved spectra of the wet-kneaded SiO_2_–MgO catalysts (Figure 4b). These CP experiments could selectively enhance surface magnesium silicate species due to more dense protons at the catalyst surface compared to bulk species, either by the physisorbed water or by surface hydroxyl groups. Notably, the ^29^Si resonances for layered magnesium silicates such as lizardite, stevensite, and talc (δ^29^Si = −93.8, −96.4, and −97.6 ppm, respectively) were greatly enhanced for WK-dried and WK-500, which means a more efficient cross-polarization transfer was evidenced. It should be clearly noted here that the enhanced intensities of CP experiments largely depend on several factors of polarization transfer from ^1^H to ^29^Si, such as the proton density around silicon nucleus, the distances between protons and silicon, and the transfer efficiency modulated by the silicon environment (i.e., different relaxation rates). For example, despite Q4 species being the most abundant in pure SiO_2_, the signal intensities of the simple silanol groups (Q3 at −100.0 ppm) are more enhanced. For WK-500, the signal intensity at −77.6 ppm, which is related to the intermediate species between forsterite and enstatite [39], was significantly enhanced. This suggests that the intermediate species between forsterite and enstatite still preserves structural hydroxyl groups due to the low temperature calcination despite their anhydrous nature; the intermediate species are hydrophilic and surrounded by considerable amounts of physisorbed water molecules. Similarly, amorphous hydrous magnesium silicates (in the region of −85 and −92 ppm) seemed to increase upon calcination (for WK-600 and WK-700) due to the transition of hydroxyl groups from ill-defined to (relatively) ordered structures [19]. Further calcination at 800 °C brought about a sharp resonance at −93.8 ppm, suggesting that the local crystallinity of lizardite increases and that transfer of magnetization from proton to silicon occurs more effectively. WK-800 displayed the broadened Q4 resonance, most probably due to the formation of additional SiO_2_ phases by the dehydroxylation of the magnesium silicates at high-temperature calcination.

For WK-900, ^29^Si resonance at −61.3 ppm was not observed with the CP experiment because of an absence of structural protons in forsterite. Notably, a new resonance at −96.4 ppm was clearly observed, attributed to a defect site containing talc-like magnesium silicate and stevensite [19,42]. Taken together with the ^29^Si DE NMR spectra for WK-900 (Figure 4a), this suggests that the dehydroxylation of structural hydroxyl groups occurring the surface of WK-900 mostly consists of forsterite, while a certain number of defect-containing stevensites decorate the grain boundary of forsterite. By the CP experiment, the broadened features of Q4 resonance, which was observed for WK-800, was hardly seen for WK-900, indicating that the SiO_2_ is now fully covered by forsterite and stevensite species. The observed ^29^Si species by ^1^H–^29^Si CP MAS NMR are summarized in Appendix A.

We used 2D ^1^H–^29^Si HETCOR NMR to study the correlation between protons and ^29^Si species for the WK-500, -800, and -900 catalysts (Figure 5). In this 2D NMR experiment, only the ^29^Si species in close proximity to protons are observed and the intensity of the cross-peaks (δ^29^Si, δ^1^H) is semiquantitative because the dipolar couplings are inversely proportional to the distance between the nuclei. For WK-500 and WK-800, we observed broad resonances at the pairs of shifts around (−100.0, 5.0) ppm (Figure 4), ascribed to physisorbed water molecules on the surface Q3 silanol groups. The observed resonance at 2.0 ppm in 1D ^1^H NMR experiment (Figure 3) was not observed in the 2D HETCOR experiment, suggesting that the proton is attributed to Mg–OH groups either on the surface of MgO or magnesium silicates, especially for the high-temperature calcined WK-900 catalyst. 

The presence of magnesium silicates is clearly evidenced in the 2D ^1^H–^29^Si HETCOR NMR spectra, with a clear correlation signal of structural protons at δ^1^H ≃ 0.7. WK-500 in the cross-peaks of the intermediate species between forsterite and enstatite at (−77.6, 0.7) and (−80.6, 0.8) ppm. After calcination at 800 °C, we observed large chemical shifts in F1(^1^H) dimension to (−73.6, 5.4) and (−81.2, 4.3) ppm, indicating that the structural protons are released from the surface of the intermediate magnesium silicates and that water is physisorbed on these sites. We speculate that the intermediate species still have oxygen-rich planes at their surfaces because of the considerable signals attributed to the adsorbed water on the surface (Figure 3 and Figure 5). In addition, the amorphous magnesium silicate and lizardite species were found to be thermally stable, displaying a slight shift from (−84.9, 0.2) and (−92.7, 0.3) to (−86.5, 1.1) and (−92.8, 1.0) ppm. 

Although talc is one of the hydrous magnesium silicates in the wet-kneaded SiO_2_–MgO catalyst [19,47], we could not observe the signal attributed to talc in the 2D HETCOR experiment, which was clearly seen in the 1D ^1^H–^29^Si CP MAS spectra of WK-dried and WK-500 (δ^29^Si = −97.6 ppm in Figure 4b). This could be due to the weak proton signals buried under the neighboring massive signals in the HETCOR experiment [48], such as the presence of a considerable amount of water on Q3 silicon species in our case. For WK-900, the resonance of forsterite was not observed in reference to the anhydrous feature, while the actual resonance of stevensite was speculated at (−96.4, 0.7) ppm. 

In summary, the formation of amorphous anhydrous magnesium silicate and lizardite is promoted by thermal treatment until 800 °C. Further calcination above 800 °C results in a phase transition at the surface of wet-kneaded SiO_2_–MgO: most of the isolated silanol groups (Q3) react with nearby magnesium species to form forsterite while maintaining a stevensite structure at the edge of the grain boundary.

### 3.2. Catalytic Activity

First, the catalytic performances of the wet-kneaded samples were compared with a physical mixture of SiO_2_ and MgO (Appendix A). Notably, the physical mixture catalyst converted ethanol mostly into acetaldehyde (77% selectivity) but almost no butadiene was observed (2.4% selectivity). This indicates that the pristine SiO_2_ and MgO have limited catalytic sites for aldol condensation, one of the suggested rate-determining step toward butadiene formation [15]. On the other hand, the wet-kneaded SiO_2_–MgO catalysts showed superior catalytic performance compared to the physical mixture catalyst in terms of ethanol conversion and butadiene selectivity. The magnesium silicates formed by wet-kneading promoted not only aldol condensation but also the sequential steps for butadiene formation.

Figure 6 shows the catalytic activity results of the prepared wet-kneaded SiO_2_–MgO catalysts for the Lebedev ethanol-to-butadiene process. The overall catalytic activities of our wet-kneaded catalysts are in line with previous results of the SiO_2_–MgO system in literature [19,49,50]. With increasing calcination temperatures, ethanol conversion decreases due to the less active sites available on the catalyst surface (Figure 6a). Interestingly, the surface-area-normalized activities (specific activities) decrease as the calcination temperature increases to 700 °C while WK-900 showed the best performance (Figure 6b). This indicates that ethanol conversion is largely influenced not only by the surface area but also by the structural properties of the distinct catalysts. More specifically, the dominant magnesium silicate phases in WK-800 (amorphous magnesium silicates and lizardite) are less active than the combination of forsterite and stevensite (WK-900) in ethanol conversion. 

Next, we performed catalytic tests at different ethanol feed rates (0.5 to 1.5 h^–1^ LHSV) to study product distribution with residence time ((Figure 6c, Appendix A). In all cases, ethanol conversion decreased with higher LHSV, meaning that the reaction rate is limited to the number of active sites (surface area) and/or the existence of competitive adsorption of ethanol and the intermediates/products. Indeed, ethanol is converted into ethylene and acetaldehyde over catalytic sites by dehydration and dehydrogenation, respectively, and the negative correlation between acetaldehyde and ethylene shows that the (limited number of) catalytic sites are occupied by those two chemicals in a competitive manner (Figure 6d). Butadiene selectivity, however, is not very sensitive to acetaldehyde although butadiene is formed via acetaldehyde.

The product selectivities of the prepared catalysts were compared at the same ethanol conversion levels (≈37%, Figure 6e). With increasing calcination temperature, the acetaldehyde selectivity decreased while ethylene selectivity greatly increased. Taken together with the characterization results, the local crystallinity of the amorphous magnesium silicates and lizardite is relative to the ethylene selectivity; that is, the protons that are strongly bound to the structures of those magnesium silicates promote the ethanol dehydration. Notably, after calcination at 900 °C, ethylene selectivity was greatly reduced to 27.5%. It is worth noting the difference in catalyst surface polarities of WK-800 and WK-900 (Figure 3 and Figure 5). Cavani et al. suggested that water that is generated in situ during the Lebedev process can transform Lewis acid sites into Brønsted acid sites, leading to an increase in ethylene selectivity [21]. In this regard, we propose that the Lewis sites of WK-800 prefers to be converted to Brønsted acid sites by in situ generated water. In addition, the rigid O–Mg–O layers on WK-900 remain nonpolar and hinder the formation of Brønsted acid sites, resulting in the notable suppression of ethylene selectivity. 

However, a detailed relationship between the structure and acid/base properties of catalysts and ethylene yield in the Lebedev process is not clarified yet [6]. For example, by pyridine adsorption followed by FTIR, we only observed Lewis acid sites for wet-kneaded SiO_2_–MgO catalysts with the absence of Brønsted acid sites, as previously reported [15,20,51] (Appendix A). However, Taifan et al. recently reported the presence of Brønsted acid sites on the surface of wet-kneaded MgO–SiO_2_ using diffuse reflectance infrared Fourier transform spectroscopy (DRIFTS) with ammonia as a probe molecule [52]. They claimed that the relatively small size of NH_3_ could penetrate small pores, while pyridine could not reach them (the kinetic diameters of ammonia and pyridine are 0.26 [53] and 0.57 nm [54], respectively). We compared the number of acid sites characterized by pyridine adsorption followed by FTIR and NH_3_–TPD (Appendix A). For WK-500, pyridine titrates fewer acid sites compared to NH_3_–TPD. This could be attributed to the less-accessible acidic sites in the interlamellar spacings for pyridine compared to ammonia. Talc is the one of the predominant phases of WK-500 (Figure 4) and the interlamellar spacings of talc are 0.1–0.6 nm depending on the degree of hydration [55,56]. There is also a possibility that the weak acidity of the silanol group on silica and/or on the magnesium silicates could not be probed by pyridine [6]. 

Previously, forsterite was considered a detrimental magnesium silicate in the Lebedev process. For example, Ochoa et al. indicated that forsterite (MgO–SiO_2_ prepared by the sol-gel method) is responsible for the ethylene yield [21]. Zhu et al. also reported that crystalline forsterite in a MgO–SiO_2_ catalyst (silica gel impregnated by magnesium nitrate hydrate) is inclined toward the formation of byproducts such as ethylene [51]. However, in our results, forsterite on a wet-kneaded SiO_2_–MgO catalyst, together with stevensite, showed not only significantly reduced ethylene formation but also greater selectivity to butadiene. In the Lebedev process, it is reported that a balance between the acidic and basic sites of the catalyst is key for increasing butadiene selectivity [15,20,57]. Szabó et al. recently showed that the catalytic activity is highly dependent on the basicity of the non-metal-promoted MgO–SiO_2_ catalysts [47]. The basicity of wet-kneaded SiO_2_–MgO catalysts was characterized using CO_2_ as a probe molecule (the kinetic diameter of CO_2_ = 0.33 nm [58]) (Appendix A). Interestingly, the CO_2_–TPD results indicated that, despite its low surface area, WK-900 exhibited a higher number of basic sites than WK-800. As the WK-900 catalyst mainly composed of forsterite and stevensite only decorates its grain boundary, the TPD result indicates that forsterite can provide adsorption sites for acidic molecules such as CO_2_. Recently, Sato et al. reported that the catalytic function of the edges (stevensite) are different from the surfaces of clay minerals such as saponite, which is a structural analogous to our stevensite/forsterite materials [59]. The catalytic performance of WK-900, especially for forsterite, thus, must be considered together with stevensite. 

Finally, to verify that the performance of catalyst materials can be directly related to the structure of the fresh samples, we performed a PXRD study of the samples after catalytic testing. The diffractograms of spent catalysts are identical to the fresh ones (Appendix A), suggesting that catalytic testing does not alter the long-range order of the materials. In addition, we monitored the catalytic conversion for 24 h time-on-stream and found that there are only 2% difference between the initial and final ethanol conversions, which demonstrates that the active sites do not change their nature or significantly deactivate in the studied time frame (Appendix A).

## 4. Conclusions

Wet-kneaded SiO_2_–MgO catalysts exhibit superior catalytic performance when compared to physical mixtures of SiO_2_ and MgO owing to the surface magnesium silicates formed during wet-kneading. An increase in the calcination temperature promotes the formation of amorphous hydrous magnesium silicates and structurally ill-defined lizardite on the surface of wet-kneaded catalysts, which are responsible for ethylene formation. Notably, forsterite, which was decorated with stevensite at the edge of grains, showed noteworthy catalytic performance for butadiene with reduced ethylene selectivity. In view of our findings, it is demonstrated that a delicate balance of different magnesium silicates is crucial for acid–base properties and the corresponding catalytic performance in the Lebedev process. The insights thus gained in the structure–activity relation for the magnesium silicates formed during catalyst preparation can enable further optimization for the selective ethanol-to-butadiene process. 

## Data Availability

The data presented in this study are available on request from the corresponding authors.

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
