# Peer review of "The Importance of Thermal Treatment on Wet-Kneaded Silica–Magnesia Catalyst and Lebedev Ethanol-to-Butadiene Process"

_nanomaterials, 2021, doi:10.3390/nano11030579_

Round 1
Reviewer 1 Report
This paper describes the synthesis of silica-magnesia catalyst by wet kneading and its application in the Lebedev process. The deep characterization of the materials, mainly the solid-state NMR studies, is very elegant. The correlation of the catalytical results with the acid-base species present in the different phases is also very interesting. In summary, the reading of this manuscript is a very inspiring experience. In my opinion, it deserves to be published in Nanomaterials with minor changes.
I think that footnote in figure 5e is wrong. It shows (e) the product distribution at similar ethanol conversions for the different catalysts under study.
It should be very useful to include table S1 in the manuscript.
Since the results are related to the acid-base properties of the materials, the authors should also consult “MgO? SiO2 Catalysts for the Ethanol to Butadiene Reaction:The Effect of Lewis Acid Promoters” DOI: 10.1002/cctc.202001007
Author Response
Reviewer 1
Review Report Form
Open Review
This paper describes the synthesis of silica-magnesia catalyst by wet kneading and its application in the Lebedev process. The deep characterization of the materials, mainly the solid-state NMR studies, is very elegant. The correlation of the catalytical results with the acid-base species present in the different phases is also very interesting. In summary, the reading of this manuscript is a very inspiring experience. In my opinion, it deserves to be published in Nanomaterials with minor changes.
We thank the reviewer for the positive assessment of the work presented. Please find the response to the minor points below.
I think that footnote in figure 5e is wrong. It shows (e) the product distribution at similar ethanol conversions for the different catalysts under study.
The footnote in Figure 5e has been revised as suggested.
It should be very useful to include table S1 in the manuscript.
We thank for the suggestion and Table S1 has now been included in the manuscript as Table 1. The table numbering has been also modified not only in the manuscript but also in the supporting information.
Since the results are related to the acid-base properties of the materials, the authors should also consult “MgO? SiO2 Catalysts for the Ethanol to Butadiene Reaction:The Effect of Lewis Acid Promoters” DOI: 10.1002/cctc.202001007
The article has been implemented in the results and discussions of the manuscript.
Szabó et al. recently showed that the catalytic activity is highly dependent on the basicity of the non-metal promoted MgO-SiO2 catalysts.[46]
Reviewer 2 Report
In this paper, the authors prepared and characterized in detail a series of wet–kneaded SiO2–MgO catalysts varying calcination temperatures.
The paper is properly divided in sections and sub-sections, but it needs some corrections before its publication on the journal.
- The authors should improve the language in the text, in order to eliminate some typing errors;
- The authors should specify in which a way they calculated the LHSV of the experimental tests;
- The authors should compare the performance of their catalysts with some other ones present in literature; in this sense, they should also extend the literature survey by adding more recent papers: some examples are 10.1021/acscatal.8b03515, 10.1016/j.fuproc.2019.04.036;
- The authors should check the acronyms, since some of them are not defined in the text;
- The authors mixed the catalyst with SiC (I think silicon carbide) in the experimental test: why this choice? Why did they not use a different material?
- Which is the duration of the experimental tests? Did the authors perform any characterization on the spent catalysts, in order to investigate if any modification occur in their structure?
- Did the authors perform any durability test in order to investigate the deactivation of the catalysts?
Round 2
Reviewer 2 Report
The authors well improved the paper